# Time-Restricted Eating in Real-World Healthcare Settings: Utilisation and Short-Term Outcomes Evaluation

**DOI:** 10.3390/nu16244426

**Published:** 2024-12-23

**Authors:** Hilmi S. Rathomi, Judith Katzenellenbogen, Nahal Mavaddat, Kirsty Woods, Sandra C. Thompson

**Affiliations:** 1School of Population and Global Health, University of Western Australia, Crawley, WA 6009, Australia; 2Faculty of Medicine, Universitas Islam Bandung, Bandung 40116, Indonesia; 3UWA Medical School, University of Western Australia, Crawley, WA 6009, Australia; 4Metabolic Health Solutions, Bentley, WA 6102, Australia; 5Western Australian Centre for Rural Health, University of Western Australia, Geraldton, WA 6530, Australia; 6School of Allied Health, University of Western Australia, Crawley, WA 6009, Australia

**Keywords:** chart review, dietary intervention, metabolic health, time-restricted eating

## Abstract

Background: Time-restricted eating (TRE) shows promise for managing weight and metabolic issues, yet its application in real-world healthcare settings remains underexplored. This study aims to assess the real-world utilisation and short-term outcomes of TRE in clinical practice. Methods: This observational study used a retrospective chart review of 271 adults who attended a metabolic specialist clinic between 2019 and 2023. Descriptive statistics and multivariable logistic regression were used to identify factors associated with TRE adoption, while paired sample *t*-tests evaluated changes in outcomes among those who received TRE advice. Results: Among the 271 patients, 76% were female, 90% Caucasian, and 94% overweight/obese. Of all patients, 47.2% received TRE advice, mainly using the 16:8 method, alongside additional dietary guidance for 60% of patients. Working status and baseline metabolic profiles were the only factors significantly associated with TRE adoption. Among those who followed TRE, 81% experienced modest but significant reductions in weight (−1.2 kg, *p* < 0.01), BMI (−0.4 kg/m^2^, *p* < 0.01), and waist circumference (−3.7 cm, *p* < 0.01). Conclusions: This study highlights TRE as a feasible and practical dietary strategy for improving metabolic health in healthcare settings. However, further research and improved data capture are needed to explore long-term adherence, potential adverse effects, and the effectiveness of TRE across diverse patient populations.

## 1. Introduction

Time-restricted eating (TRE) has garnered attention as a compelling dietary strategy due to its potential metabolic benefits [1,2,3,4]. This approach involves restricting food intake to specific time windows, typically within 6–12 h each day, while fasting for the remainder [3,5]. Emerging evidence suggests that TRE may influence various metabolic parameters, including weight management, glucose regulation, and lipid metabolism, thus holding promising implications for managing metabolic disorders such as obesity, type 2 diabetes, and cardiovascular disease [6,7]. The mechanism behind TRE is thought to involve extended fasting periods that promote fat burning and align eating patterns with the body’s circadian rhythms, thereby optimising metabolism [3,8].

While research on the efficacy of TRE in controlled settings continues to expand, a gap remains in understanding its utilisation and effectiveness in real-world clinical practice. Previous studies have explored TRE in real-world settings, but mostly within community contexts [9,10]. Although some research has examined its use in healthcare settings like rural clinics and primary care, TRE has often been part of broader lifestyle interventions rather than a primary focus [11,12]. Implementing TRE in clinical practice presents unique challenges and considerations, including variations in healthcare provider approaches, patient characteristics, and adherence to dietary recommendations. Investigating the utilisation and outcomes of TRE in healthcare settings is critical for informing evidence-based practice and improving patient care [13]. Understanding the factors that influence its use in real-world settings will help healthcare providers tailor dietary interventions to better support metabolic health.

Healthcare settings offer a promising avenue for delivering health interventions like TRE, as they provide personalised advice based on patients’ individual conditions and preferences [12,14]. Clinics also serve as an ideal setting for evaluating intervention outcomes through routine metabolic measurements taken during patient visits.

This study aims to explore the utilisation of TRE in a real-world healthcare setting through a retrospective chart review. Rather than seeking to establish causality or evaluate long-term effects, the primary focus is to understand how TRE is integrated into clinical practice. By analysing electronic health records (EHR) of patients who received TRE advice during their visits, we assess the prevalence and patterns of TRE adoption among patients with diverse sociodemographic and medical profiles. Additionally, we investigate whether specific patient characteristics are associated with a higher likelihood of adopting TRE. As a secondary objective, we examine short-term changes in metabolic outcomes, including weight, BMI, and waist circumference, among those who adopted TRE.

## 2. Methods

### 2.1. Study Design and Setting

This before-and-after observational study employed a retrospective chart review, gathering data from EHR to assess the implementation and outcomes of TRE. The data were sourced from two specialised metabolic health allied health clinics in Perth, Western Australia. These clinics utilised an ECAL^®^ machine for indirect calorimetry testing [15] to assess patient metabolism profiles. Clinicians were qualified Exercise Physiologists or Dietitians who provided face-to-face services primarily focusing on weight loss and associated health conditions, with patients incurring out-of-pocket expenses for professional consultations and testing. Systematic data collection commenced in 2019 when the clinic transitioned to online data collection.

### 2.2. Participant Selection

This study comprised two stages of analysis involving different cohorts. Stage 1, referred to as the ‘baseline cohort analysis’, included data from adults (aged ≥18 years) who attended the clinic between 2019 and mid-2023, provided informed consent for research use of their data, and had key data points available. In Stage 2, referred to as the ‘TRE cohort analysis’, we included all patients who received TRE advice in order to describe TRE utilisation among patients. Only TRE patients with at least two visits to the clinic were included for outcomes evaluation.

### 2.3. Clinic Protocols

#### 2.3.1. Prior to Clinic Visit

Before their initial clinic visit and as per clinic protocol, patients completed an online self-report questionnaire covering personal information, health history, lifestyle, and symptoms such as fatigue, energy swings, sweet cravings, and any difficulty losing weight. Patients also received instructions to ensure they met the requirements for metabolic testing, which included fasting for at least 4 h.

#### 2.3.2. First Visit Assessment

Upon arrival, patients had a consultation with an exercise physiologist or dietitian who reviewed their case history and focused on key areas highlighted in their pre-test questionnaire. Following the consultation, anthropometric measurements—such as weight, height, and waist circumference—were taken. Indirect calorimetry (ECAL^®^) was then used to measure metabolic parameters, including resting energy expenditure and fuel utilisation (glucose or fat) [16]. The primary outcome from the indirect calorimetry test is the respiratory quotient (RQ), which indicates the ratio of glucose to fat used for metabolism and reflects the patient’s metabolic flexibility [15,16,17]. A lower RQ indicates a shift toward greater fat oxidation [16]. Although there is no consensus on the ideal RQ level, the clinic established an internal reference range of 0.75 to 0.85 as “normal”. Patients with an RQ value below 0.85, corresponding to a fat oxidation level greater than 50%, were considered to have a “good” RQ. These individuals, termed “fat burners”, demonstrated an enhanced ability to use fat as their primary energy source. In addition, body composition measurements were performed as deemed necessary by the clinician.

#### 2.3.3. Intervention

All information gathered during the initial visit assessment was used to develop personalised recommendations for diet, exercise, and other lifestyle modifications for individual patients. The clinic prioritised strategies to reduce the insulin response of patients’ diets, aiming to improve metabolic flexibility and fat oxidation, which in turn supported weight loss and overall health. Common recommendations included therapeutic carbohydrate restriction (TCR), increased intake of whole foods, healthy fats, and non-starchy vegetables, as well as adjustments to meal timing, such as intermittent fasting or TRE. Exercise, specific supplements, and other corrective measures were also recommended by the clinicians when necessary [18].

At the first visit, TRE was typically recommended for patients with a good RQ (“fat burners”) as these patients were believed to have a greater ability to utilise body fat as their primary fuel source. This fat-burning capacity was considered beneficial for reducing hunger and, in turn, promoting adherence to TRE, which involves prolonged periods of fasting. Those who were prescribed TRE at the first visit are referred to as ‘First-visit TRE’. Conversely, patients with suboptimal RQ (>0.85) commonly received different recommendations, often focusing initially on TCR to enhance their metabolic flexibility [18].

All clinicians followed general clinic guidelines for recommending TRE to patients who had favourable RQs. However, treatment plans were developed collaboratively through negotiation between the clinician and the patient to ensure that they were acceptable and sustainable for the individual. Consequently, there were no rigid treatment pathways, and some variation in the delivery of TRE advice occurred based on patient preferences and clinician discretion. The interventions, as agreed between clinician and patient, were documented in the clinical notes. Each visit concluded with a follow-up consultation, incorporating motivational techniques to encourage adherence to the recommended lifestyle changes and to plan future visits.

#### 2.3.4. Follow-Up Visits

At each follow-up visit, indirect calorimetry testing was conducted, with additional anthropometry and body composition assessments performed at the discretion of clinicians. The advice provided, including for taking up TRE during follow-up visits, depended on individual progress and evaluations including indirect calorimetry findings. While TRE can be recommended by clinicians to patients at the first visit (‘First-visit TRE’) depending on their RQ, TRE was frequently advised at the second visit (‘Second-visit TRE’) or at a subsequent visit (‘Subsequent TRE’) when an optimal RQ reading was obtained, or when patients requested alternative approaches to help with weight loss.

The time intervals between visits and interventions were flexible and tailored to each patient based on their results, practitioner insights, and scheduling. For further details on the patient’s pathway in the clinic, please see Appendix A.

### 2.4. Data Collection

Data for the Stage 1 cohort were extracted from EHR using a standardised data collection form, incorporating variables of interest. These variables included sociodemographic characteristics, baseline health conditions, and psychosocial practices obtained from self-report questionnaires. Metabolic measurements included respiratory quotient, fat oxidation level, weight, waist circumference, and body fat percentage.

Information regarding the interventions or advice to the patients was extracted from clinical notes, as these detailed the interventions agreed upon between patients and clinicians. These data were then categorised according to the types of interventions received, aligning with the study objectives. Categories included TRE, dietary modifications and supplements, exercise, and other lifestyle factors (including sleep and relaxation). Reducing carbohydrate intake was categorised separately from dietary modifications, as it is a general recommendation based on the clinic’s guidelines [18] and is commonly recommended together with TRE in some clinical settings [11,12].

Outcomes evaluated in Stage 2 included changes in weight, BMI, waist circumference, body fat, and fat oxidation levels, comparing parameters measured in the TRE initiation visit and the follow-up visit. See Figure 1 for further details.

### 2.5. Statistical Analysis

In the Stage 1 cohort analysis, we employed descriptive statistics, including means, medians, percentages, and confidence intervals, to characterise the study population and baseline measurement. Demographic data were summarised using mean (standard deviation) for parametric data and median (interquartile range) for non-parametric data. Categorical data were presented as counts (percentages). To analyse the relationship between baseline data and TRE utilisation across groups, we used bivariate analysis with appropriate tests (one-way ANOVA, chi-square). Statistical significance was determined by a *p*-value <0.05. Additionally, multivariable logistic regression was utilised to examine the association of covariates with TRE utilisation, adjusting for age, gender, medical conditions, and RQ. RQ was included in the adjusted model as it is the major consideration for clinicians when prescribing TRE and was also significant in our analysis (refer to Appendix A). No clinician-specific data was included in the analysis due to the absence of records linking individual patients to specific clinicians.

For the Stage 2 cohort analysis, we designated the patient’s first visit to the clinic as the “baseline”. The visit during which TRE was introduced to patients was termed the “TRE initiation” visit. Subsequent visits after the initiation, for those advised on TRE, were termed “TRE follow-up” visits.

For the evaluation of outcomes in the Stage 2 cohort, paired *t*-tests were used to compare changes in weight, BMI, waist circumference, and fat oxidation from TRE initiation to the subsequent follow-up visit. To account for multiple testing, a Bonferroni correction was applied, setting the level of statistical significance at 0.01 (0.05/5 tests). Furthermore, multivariable linear regression was used to examine the difference in the outcomes adjusted for follow-up duration. All statistical analyses were conducted using STATA 13 (StataCorp, LP, College Station, TX, USA).

### 2.6. Ethics

This study received approval from the UWA Human Research Ethics Committee (HREC) with project number 2021/ET000993. Patient confidentiality was strictly maintained throughout the data collection and analysis process. Only de-identified data were used for analysis, with any potentially identifying information securely handled and stored by the clinic data manager. All participants provided written informed consent to the clinic for their data being used for research purposes.

## 3. Results

Data were available from the clinic for 373 patients. After excluding those with incomplete measurements, non-adults, and those for whom the advice given could not be identified, 271 patients were eligible for Stage 1 analysis. Of those, 128 were advised to follow TRE, and 97 of them had follow-up data available for before-and-after comparison of outcomes (Figure 1).

In terms of the distribution of TRE introduction timings, 50 patients (39%) were classified as “First-visit TRE”, 61 (48%) started TRE during their second visit (classified as “Second-visit TRE”), and the remaining 13% comprised the “Subsequent TRE”. The interval between initiation visits and follow-up visits varied, with a median of 30 days (IQR 21, 49).

### 3.1. Baseline Cohort

Among patients in the Stage 1 Cohort, the mean age was 47.4 years (SD = 12.6), with the majority being female (*n* = 207, 76%) and of Caucasian ethnicity (90%). Most were employed either full-time (≥40 h/week) (*n* = 69, 25%) or part-time (*n* = 157, 58%), while the remainder were not working. The proportion of self-reported medical problems varied among patients, including diabetes (21%), cardiovascular problems (30%), and fatty liver (17%) (Table 1). Approximately 69% of the patients were on medications related to their medical conditions, including antihypertensives (e.g., candesartan, amlodipine), antidiabetics (e.g., metformin), lipid-lowering agents (e.g., atorvastatin, rosuvastatin), thyroid hormone replacement (e.g., thyroxine), and antidepressants (e.g., sertraline, escitalopram). Almost all patients (97%) visited the clinic for weight loss, with 68% reporting weight gain over the previous year. A minority had specific dietary preferences prior to the clinic visit, such as following a ketogenic diet (11%) or being vegan/vegetarian (3%).

At baseline, most patients were overweight (22%) or obese (72%). The mean BMI and body fat percentage were 34.1 kg/m^2^ (SD = 6.8) and 38.8% (SD = 7.6), respectively. Measurement from indirect calorimetry showed a mean RQ among patients of 0.89 (SD = 0.11) with a median fat oxidation level of 38% (IQR = 15–59). Thus, 28% of patients were classified as good fat burners based on an RQ level < 0.85 at baseline.

There were no significant differences in most characteristics and baseline measurements between non-TRE and TRE groups across different timings of introduction. Small differences were observed in age distribution, working status, prevalence of medical conditions, weight, and BMI between groups, but these differences were not statistically significant. The only significant difference observed was in calorimetry measurements indicated by RQ, fat oxidation level, and the proportion of patients identified as fat burners, with a significantly higher proportion in the “First-visit TRE” group (Table 1).

We found a similar proportion of TRE use among patients across ages, gender, working hours, habits, and medical conditions. Among all plausible factors included in the analysis, only working status was significantly associated with TRE use, with employed patients—either part-time or full-time—more likely to use TRE. There was no significant difference in the use of TRE by diabetic status, although TRE use was non-significantly lower in those with diabetes (39% vs. 49%, OR 0.65, 95% CI: 0.36, 1.17, *p* = 0.152). Although the difference was not significant, a higher proportion of TRE use was observed among those who reported routinely eating breakfast in the baseline (51% vs. 35%, OR 1.89, 95% CI 0.90, 3.94) (Table 2).

The multivariable logistic regression found that baseline RQ emerged as the only significant factor associated with TRE use (aOR = 23.02, 95% CI [8.79, 60.25], *p* = 0.000). A complete model analysis of TRE use determinants is shown in Appendix A. We used baseline RQ and all plausible factors to adjust the association between each sociodemographic factor and TRE use (Table 2).

### 3.2. TRE Cohort

In our TRE cohort (Stage 2), almost all patients included were prescribed the 16:8 TRE method (*n* = 112, 88%), although three patients opted for the One Meal a Day (OMAD) type with a small eating window. Additional dietary advice, especially carbohydrate restriction, was commonly provided, especially among “Second-visit” and “Subsequent” TRE groups, but less commonly among “First-visit TRE” (Table 3).

There was a statistically significant decrease in mean weight of −1.2 kg (95% CI = −1.5, −0.9, *p* = 0.000) and mean BMI by −0.4 kg/m2 (95% CI = −0.5, −0.3) between TRE initiation visit and the follow-up visit. Similarly, among those with waist circumference recorded at two visits, we found a significant decrease from 107 cm (SD = 14) to 104 (SD = 14). No significant changes in fat oxidation levels were evident between visits (Table 4).

When classified based on the timing of TRE introduction, we found a similar effect over different groups. However, there was a significant difference in the initial fat oxidation level between those in the “First-visit” TRE and the “Second-visit” TRE group, with the latter showing a lower fat oxidation level (68.6% ± 14 vs. 53.4% ± 23.1, *p* = 0.003).

Among patients who had complete initial and follow-up data recorded, 81% experienced weight loss, and the proportion varied slightly between the different timings of TRE introduction. The highest proportion of patients experiencing weight loss was in the “Subsequent TRE” group (87%), although the time between visits in that group was also the longest (64 days, IQR = 42–113). Some individuals across different groups (13–20%) experienced weight gain (Figure 2).

## 4. Discussion

This retrospective chart review provides valuable insights into the utilisation and short-term effects of TRE in a real-world healthcare setting focused on metabolic health. Approximately 50% of clinic attendees were advised to follow TRE, with varying timings of introduction and predominantly using the 16:8 method. Among these patients, significant improvements were observed in key metabolic parameters, including reductions in weight, BMI, and waist circumference.

The modest but significant reductions in weight, BMI, and waist circumference observed among patients advised to follow TRE in our studied clinic align with previous research demonstrating TRE’s effectiveness in promoting weight loss and reducing central adiposity [7,10]. However, the short duration of follow-up in this study highlights the need for longer-term evaluation, as weight regain is often reported after any weight loss interventions. Furthermore, the observed improvements may not be solely attributed to TRE, as factors such as dietary changes and increased physical activity, which were not measured, may have contributed.

A recent randomised controlled trial by Quist et al. found no significant weight loss from TRE compared to a control group within three months [19]. However, other studies and reviews have consistently shown TRE to have a favourable impact on weight [4,6,7]. Differences in outcomes may be explained by variations in study design, including the slight difference in eating duration between groups (around 2 h) and higher calorie intake in the TRE group in Quist’s study [19]. Given the challenges of sustained calorie reduction that often lead to weight regain [2], simple approaches with higher adherence, such as TRE [20], may be beneficial for patients to consider. Additionally, research has shown that TRE often leads to an inadvertent calorie reduction of 200–500 calories [4], or around 10–30% [21], thereby contributing to improved metabolic processes and weight loss. Moreover, a previous study demonstrated significant weight loss with TRE over a 12-month period, despite a slight regain after the peak weight loss phase (around 3–6 months) [22].

Our study also found a similar proportion of TRE use across various patient demographics and baseline characteristics, with RQ levels influencing the advice on TRE, in line with the clinic’s approach. This supports previous research indicating that both healthcare providers and patients are open to exploring alternative dietary strategies, such as TRE or intermittent fasting, for managing weight and metabolic health [4,23,24]. Further, patients are particularly drawn to TRE due to its flexibility since it can be tailored to fit their personal lifestyle and preferences, making it a more sustainable option in the longer term for many individuals [25].

The variations in the timing of TRE introduction observed in this study reflect the personalised approach adopted by clinicians, who considered individual metabolic profiles and patient readiness when recommending interventions. Patients with RQ levels indicating the presence of metabolic flexibility or an ability to utilise body fat as a primary energy source were more likely to be advised to use TRE [17]. These criteria were more often applied at the initial visit, with less emphasis placed on them in subsequent visits, indicating a pragmatic approach commonly used in real-world clinical practice. However, given the lack of consensus on the optimal RQ cut-off values and limited use of indirect calorimetry in clinical settings [26], further research is needed to assess whether RQ-based selection is an effective predictor of TRE responsiveness.

Patients with sub-optimal RQ levels in this clinic were typically advised to initially reduce their carbohydrate intake to improve fat oxidation levels prior to receiving advice to use TRE [18]. Additionally, exercise was not emphasised as a supplement to TRE in this setting, with a reduced emphasis on advising exercise stemming from the clinicians’ recognition that exercise might increase patient hunger [27] and have less impact on weight loss compared to dietary changes [28]. Many patients were already engaged in some form of exercise prior to attending the clinic.

Although no significant changes in fat oxidation levels were observed among patients prescribed TRE, indicating that TRE may not directly impact fuel utilisation, most patients started TRE with adequate fat oxidation levels. Previous research has shown that higher fat oxidation levels are not required to maintain continuing weight loss if they have been satisfactory [18]. Given the limited data on body fat percentage in this study, further research is needed to assess TRE’s direct impact on body composition. Nonetheless, meta-analyses have demonstrated that TRE facilitates fat mass reduction while preserving lean body mass [7], which is essential for metabolic health.

Our findings suggest that TRE is a practical and effective strategy for managing weight and metabolic health in real-world healthcare settings. Understanding the factors influencing TRE use and its effects on metabolic parameters in community clinics can inform the development of clinical guidelines for implementing TRE as a therapeutic strategy for specific metabolic issues. As TRE gains popularity [4], more real-world studies are needed to complement findings from controlled trials, particularly those exploring patient and practitioner perspectives, adherence, adverse effects, and long-term efficacy. It is crucial to further investigate the safety, adherence, and effectiveness of TRE in diverse patient populations. Healthcare providers may consider integrating TRE into their clinical practice, particularly for patients with obesity, metabolic problems, and related comorbidities, using a patient-centred approach that accounts for individual preferences and conditions.

### Strength and Limitations

This study presents several strengths that enrich the existing evidence on the practical application of TRE in real-world healthcare settings. A key strength lies in its use of real-world data, encompassing a diverse range of patient types, which enhances the generalisability of the findings to broader clinical contexts. Unlike controlled trials, this study provides valuable insights into how TRE is applied in routine clinical practice, where tightly controlled conditions are often impractical. By capturing data from actual patient care, the study demonstrates the feasibility of integrating TRE into clinical care across diverse patient demographics, with potential short-term metabolic benefits. Furthermore, the use of indirect calorimetry to assess metabolic flexibility and fat oxidation adds an additional layer of depth, offering a more nuanced understanding of patient responsiveness to TRE.

However, several limitations must be acknowledged. First, while the use of real-world data enhances the generalisability of findings, the potential variability in the delivery of TRE advice among clinicians may influence the interpretation of results. Although the relatively uniform approach guided by clinic protocols helps mitigate this concern, future studies should consider tracking clinician-specific data to better evaluate the impact of practitioner-related factors on the implementation of TRE.

Second, the study primarily focuses on short-term outcomes, leaving the long-term impact of TRE unassessed. Patient adherence to the prescribed intervention was not rigorously monitored, which is a common limitation in clinical practice due to time constraints [29,30].

Additionally, other potential confounding variables, such as changes in diet, physical activity, or lifestyle factors, were not systematically controlled, which may have influenced the observed metabolic improvements. Limited data on related parameters, such as lipid profiles and glucose levels, further restricted the scope of this study. The use of RQ as a criterion for the selection of patients for TRE intervention also presents limitations, as it remains relatively untested as a predictor of responsiveness to clinical interventions, including TRE, and there is a lack of consensus on optimal values for this purpose [16].

Future research should evaluate the long-term impacts of TRE, monitor adherence, address confounding variables alongside broader metabolic parameters, as well as validate RQ as a reliable tool for guiding dietary recommendations. Exploring potential adverse effects, such as hunger or headaches [4], and how to manage them could enhance TRE’s implementation and efficacy. While previous studies [9,31] and a systematic review suggest generally high adherence to TRE [20], further research with improved study designs is needed to confirm TRE’s acceptability and effectiveness across diverse patient populations.

## 5. Conclusions

Our findings support the promising potential of TRE as an acceptable and effective dietary intervention for assisting weight management and enhancing metabolic health in clinical practice. By elucidating its use and demonstrating significant improvements in metabolic profiles among patients, this study contributes to the growing body of evidence supporting the incorporation of TRE as an option within comprehensive weight and metabolic health management strategies. While no formal guidelines currently incorporate TRE as a practice, practitioners might consider it as a complementary approach to enhance metabolic improvements, particularly when combined with currently recommended strategies such as healthier eating choices and exercise. Further research in real-world contexts is necessary to optimise the implementation of TRE, examine the long-term effects of TRE, and improve patient outcomes.

## Figures and Tables

**Figure 1 nutrients-16-04426-f001:**
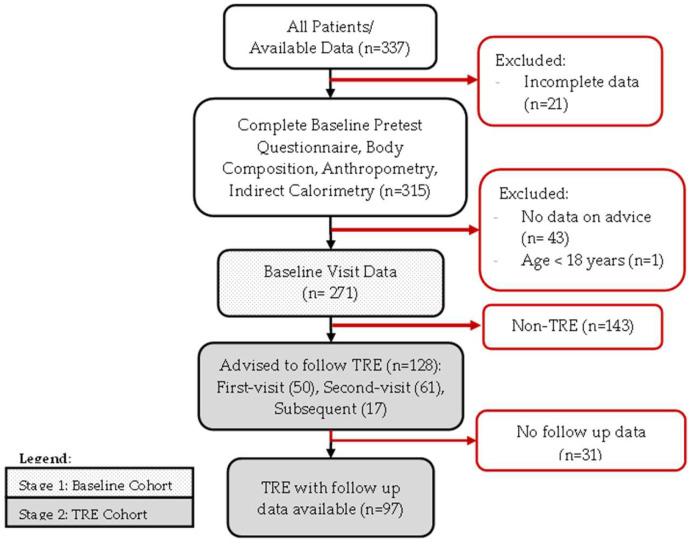
Flowchart of participants included in the study.

**Figure 2 nutrients-16-04426-f002:**
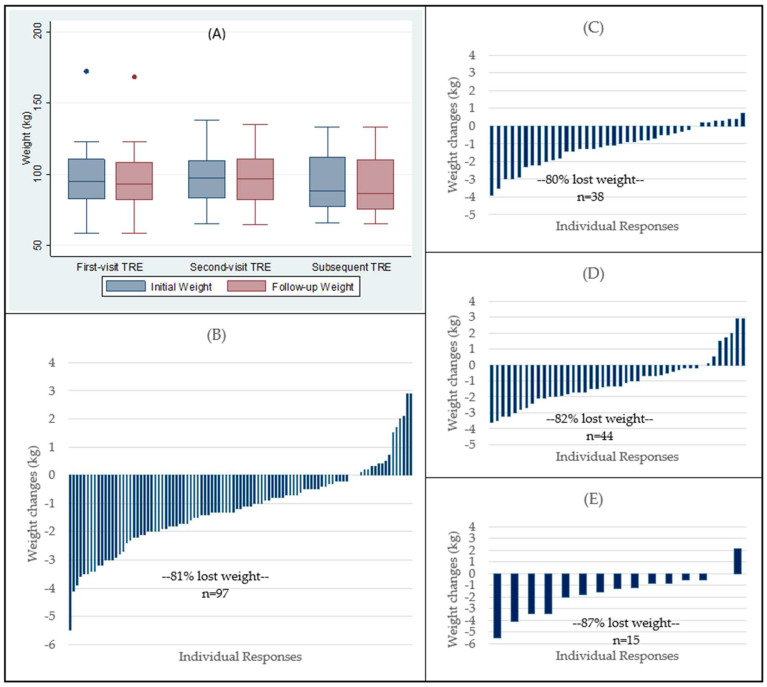
Weight changes among TRE users: Comparison of average initial and follow-up weight between different timing of TRE introduction, with dots representing outlier values (**A**), and individual changes among all TRE patients (**B**), First-visit TRE (**C**), Second-visit TRE (**D**), and Subsequent TRE (**E**).

**Table 1 nutrients-16-04426-t001:** Baseline characteristics of patients (*n* = 271).

Characteristic	All Patients (*n* = 271)	Non-TRE (*n* = 143)	First-Visit TRE (*n* = 50)	Second-Visit TRE (*n* = 61)	Subsequent TRE (*n* = 17)
Mean Age, years (SD)	47.4 (12.6)	46.5 (13.3)	50.4 (10.4)	47.7 (11.4)	45.5 (14.9)
Gender, *n* Female (%)	207 (76)	113 (79)	35 (70)	45 (74)	14 (82)
Ethnicity, *n* Caucasia*n* (%)	245 (90)	125 (87)	45 (90)	59 (97)	16 (94)
Working Status, *n* (%)					
Retired	46 (17)	32 (22)	5 (10)	6 (10)	3 (18)
<40 h/week	156 (58)	77 (54)	28 (56)	39 (64)	12 (71)
≥40 h/week	69 (25)	34 (23)	17 (34)	16 (26)	2 (12)
Medical Problems, *n* (%)					
Diabetes	57 (21)	35 (24)	7 (14)	13 (21)	2 (12)
Cardiovascular problem	80 (30)	42 (30)	17 (34)	16 (26)	5 (29)
Fatty liver	46 (17)	29 (20)	5 (10)	11 (18)	1 (6)
PCOS (% from female)	26 (13)	14 (12)	2 (6)	8 (18)	2 (14)
On medication	187 (69)	102 (71)	37 (74)	37 (61)	11 (65)
Bariatric Surgery	11 (4)	8 (6)	2 (4)	1 (2)	0 (0)
Weight gain over the past year, *n* (%)	185 (68)	97 (67)	37 (74)	41 (67)	10 (59)
Coming for weight loss, *n* (%)	263 (97)	138 (96)	48 (96)	60 (98)	17 (100)
Food preference, *n* (%)					
Low carb/keto	31 (11)	17 (12)	8 (16)	5 (8)	1 (6)
vegan/vegetarian	8 (3)	4 (3)	0 (0)	4 (7)	0 (0)
Eating habits, *n* (%)					
Eat breakfast	177 (66)	87 (61)	33 (66)	45 (74)	12 (71)
Skipping meals	72 (26)	40 (28)	12 (24)	15 (25)	5 (29)
BMI category, *n* (%)					
Normal-underweight (<25 kg/m^2^)	16 (6)	9 (6)	6 (12)	1 (2)	0 (0)
Overweight (>25 kg/m^2^)	59 (22)	33 (23)	9 (18)	10 (16)	7 (41)
Obese (>30 kg/m^2^)	196 (72)	101 (70)	35 (70)	50 (82)	10 (59)
BMI, kg/m^2^ (SD)	34.1 (6.8)	34.1 (7.5)	33.5 (6.7)	34.5 (5.4)	34.3 (5.7)
Weight, kg (SD)	96.6 (21.8)	96.3 (23.8)	94.5 (20.2)	98.7 (18.4)	97.1 (19.8)
Waist Circumference, cm (SD)	105 (17)	104 (18)	105 (15)	107 (18)	106 (13)
Bodyfat Percentage, % (SD)	38.8 (7.6)	38.7 (7.6)	38.6 (7.8)	39.4 (7.8)	37.7 (7.4)
Fat Burner, *n* (%)	104 (38)	38 (26)	44 (88) *	15 (25)	7 (41)
Respiratory Quotient, RQ (SD)	0.89 (0.11)	0.91 (0.1)	0.8 (0.08) *	0.92 (0.11)	0.91 (0.11)
Fat Oxidation Level (%), median (IQR)	38 (15–59)	35 (8–53)	67.5 (59–76) *	35 (15–45)	45 (5–59)

Note: * *p* < 0.05 indicates a statistically significant difference compared to the non-TRE group. “First-visit TRE” were introduced to TRE during their first visit, “Second-visit TRE” during their second visit, “Subsequent TRE” after their third visit. PCOS: polycystic ovary syndrome, BMI: body mass index, RQ: respiratory quotient, Fat Burner: patients with RQ < 0.85.

**Table 2 nutrients-16-04426-t002:** Univariate relationships of patients’ baseline characteristics with TRE.

Variable	% of TRE Users	Crude OR (95% CI)	*p*-Value	aOR (95% CI) ^	*p*-Value
Gender					
Male	53%	.ref		.ref	
Female	45%	0.73 (0.42, 1.28)	0.281	0.99 (0.36, 2.71)	0.98
Age					
Early adult (<45 years)	42%	.ref		.ref	
Mid adult-elderly (≥45 years)	50%	1.41 (0.86, 2.31)	0.169	1.60 (0.65, 3.94)	0.309
Working status					
Retired/Not working	30%	.ref		.ref	
<40 h/week	50%	2.34 (1.16, 4.73)	0.017 *	4.8 (1.42, 13.48)	0.010 *
>40 h/week	51%	2.35 (1.07, 5.16)	0.033 *	2.76 (0.82, 9.27)	0.102
Medication use					
No	51%	.ref		.ref	
Yes	45%	0.79 (0.47, 1.33)	0.382	0.55 (0.22, 1.36)	0.199
Cardiovascular Problem					
No	47%	.ref		.ref	
Yes	47%	1.01 (0.60, 1.71)	0.954	1.66 (6.67, 4.10)	0.270
Diabetes					
No	49%	.ref		.ref	
Yes	39%	0.64 (0.35, 1.16)	0.143	0.42 (0.14, 1.23)	0.113
Skipping meals habit					
Never-Rare	50%	.ref		.ref	
Most-Every Day	44%	0.78 (0.41, 1.49)	0.450	1.14 (0.39, 3.37)	0.805
Breakfast habit					
Never-Rare	35%	.ref		.ref	
Most-Every Day	51%	1.90 (0.91, 3.99)	0.085	2.54 (0.69, 9.07)	0.163
Initial BMI					
Not overweight	44%	.ref		.ref	
Overweight	44%	1.01 (0.33, 3.08)	0.982	2.49 (0.43, 14.32)	0.306
Obese	48%	1.21 (0.43, 3.38)	0.717	1.60 (0.33, 7.83)	0.561

^ adjusted with all sociodemographic and baseline respiratory quotient (RQ), * *p* < 0.05 indicates a statistically significant difference.

**Table 3 nutrients-16-04426-t003:** Profile of interventions utilised among patients advised to follow TRE (*n* = 128).

TRE Regimen	First-Visit TRE (*n* = 50)	Second-Visit TRE (*n* = 61)	Subsequent TRE (*n* = 17)	All TRE (*n* = 128)
Type of TRE, *n* (%)				
16:8	43 (86)	53 (87)	15 (88)	112 (88)
eTRE	3 (6)	5 (8)	1 (6)	9 (7)
OMAD	1 (2)	1 (2)	1 (6)	3 (2)
Not specified	3 (6)	2 (3)	0 (0)	5 (4)
Additional Intervention, *n* (%)				
Low/zero carbs	11 (22)	57 (93)	16 (94)	84 (66)
Other dietary and supplement	12 (24)	13 (21)	7 (41)	43 (34)
Exercise and other lifestyles	3 (6)	4 (7)	0 (0)	7 (5)

Note: eTRE: early Time-Restricted Eating, OMAD: One Meal a Day.

**Table 4 nutrients-16-04426-t004:** Metabolic profile changes from introduction of TRE to next follow-up visit among those who were advised to follow TRE (*n* = 97), by TRE timing.

Parameters	Mean (SD) at Initiation	Mean (SD) at Follow Up	Mean Difference (95% CI)	*p*-Value	%Changes
All TRE (median follow-up duration [IQR]: 30 [21, 49] days)					
Weight (kg)	97.1 (19.2)	95.9 (19.1)	−1.2 (−1.5, −0.9)	0.000 *	−1.2%
BMI (kg/m^2^)	34.4 (5.8)	33.9 (5.7)	−0.4 (−0.5, −0.3)	0.000 *	1.2%
Waist Circumference (cm) (*n* = 16)	107 (14)	104 (14)	−3.7 (−6.1, −1.2)	0.003 *	−3.5%
Fat Oxidatio*n* (%)	60.1 (21.2)	59.7 (22.7)	−0.3 (−5.5, 4.8)	0.45	−0.5%
First-visit TRE (median follow-up duration [IQR]: 15 [14, 23] days)					
Weight (kg)	96.8 (19.9)	95.6 (19.5)	−1.1 (−1.5, −0.8)	0.000 *	−1.1%
BMI (kg/m^2^)	34.4 (6.5)	34.1 (6.4)	−0.4 (−0.5, −0.3)	0.000 *	−1.1%
Waist Circumference (cm) (*n* = 11)	112 (14)	107 (15)	−4.2 (−7.9, −0.5)	0.015 *	−3.8%
Fat Oxidatio*n* (%)	68.6 (14)	63,5 (22.6)	−5.1 (−4.7, 14.9)	0.149	7.4%
Second-visit TRE (median follow-up duration [IQR]: 41 [28, 53] days)					
Weight (kg)	98.6 (18.6)	97.6 (18.6)	−1.0 (−1.5, −0.6)	0.000 *	−1.0%
BMI (kg/m^2^)	34.7 (5.3)	34.4(5.3)	−0.4 (−0.5, −0.2)	0.000 *	−1.1%
Waist Circumference (cm) (*n* = 2)	90 (5.6)	88 (7.1)	−2 (−14, 10)	0.147	−2.2%
Fat Oxidatio*n* (%)	53.4 (23.1)	55.8 (23.0)	2.4 (−4.9, 9.7)	0.741	4.5%
Subsequent TRE (median follow-up duration [IQR]: 64 [42, 113] days)					
Weight (kg)	93.8 (20.0)	92.1 (20.2)	−1.7 (−2.7, −0.6)	0.002 *	−1.8%
BMI (kg/m^2^)	33.4 (5.3)	32.8 (5.2)	−0.6 (−0.9, −0.3)	0.001 *	−1.8%
Waist Circumference (cm) (*n* = 3)	103 (3.6)	100 (3.2)	−2.7 (−4.1, −1.2)	0.008 *	−2.6%
Fat Oxidatio*n* (%)	58.1 (24.2)	62.1 (21.2)	3.9 (−5.6, 13.5)	0.804	6.7%

Note: * *p* < 0.05 indicates statistical significance.

## Data Availability

The data supporting this study’s findings are available from the corresponding author upon reasonable request.

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
