# Peer review of "Time-Restricted Eating in Real-World Healthcare Settings: Utilisation and Short-Term Outcomes Evaluation"

_nutrients, 2024, doi:10.3390/nu16244426_

Round 1

Reviewer 1 Report

Comments and Suggestions for Authors

I just have a few minor comments:

- Please clarify in Table 2: “Early adulthood” and “Middle adulthood - old age”, what does this mean, so what is the group classification?

- what kind of Medication patients used? For example, did they use any light boosters / metabolism accelerators? This should be added / explein in the text

- What were the ranges of ''intermittent fasting''? What percentage of people used them?

- With information about health status and health problems (e.g. CVD), have patients' rates improved? Has this been studied at all? E.g. measurement of blood pressure, blood lipid profile, glucose?

- In Table 4, add the time of follow-up in the column (e.g., after 4, 6 months)

- as economic status asked when collecting the interview? Because it may have deteriorated or improved, which affects the level of consumption of selected products (please add in text)

- The conclusions are very general, I would rather detail on a few points , especially since the authors mention improving metabolic health (that is who) and adding information of future interventions

- I have to admit that the work has potential, but I am very interested in this particular interval of ''fasting time'' because it raises a lot of controversy. I would like to ask you to clarify this, especially since from the title I expected just to be more detailed and better described

Author Response

Comment 1: Please clarify in Table 2: “Early adulthood” and “Middle adulthood - old age”, what does this mean, so what is the group classification?

Response 1: Thank you for your comment. We have clarified the classification by adding the actual age ranges in Table 2: “Early adult” refers to ages <45 years (18–44), and “Mid adult-elderly” refers to ages ≥45 years (Page 7).

Comment 2: What kind of Medication patients used? For example, did they use any light boosters / metabolism accelerators? This should be added / explain in the text.

Response 2: We have included a description of the medications used by patients in the results section: “Approximately 69% of the patients were on medications related to their medical conditions, including antihypertensives (e.g., candesartan, amlodipine), antidiabetics (e.g., metformin), lipid-lowering agents (e.g., atorvastatin, rosuvastatin), thyroid hormone replacement (e.g., thyroxine), and antidepressants (e.g., sertraline, escitalopram)(lines 219-222).

We did not identify any metabolism boosters / accelerators used among patients.

Comment 3: What were the ranges of ''intermittent fasting''? What percentage of people used them?

Response 3: Details about the range of intermittent fasting/TRE regimens (e.g., 16:8, OMAD) and the proportion of patients following each are provided in Table 3 (Page 8). Specifically, 88% used the 16:8 method, 2% followed OMAD, 7% practiced early TRE (eTRE), and 4% had unspecified types recorded in the clinical notes. eTRE refers to an eating window starting in the morning and ending in the afternoon, as opposed to late TRE, where the window begins in the afternoon and ends in the evening.

We included in the results: “almost all patients included were prescribed the 16:8 TRE method (n=112, 88%), although three patients opted for the One Meal a Day (OMAD) type with a small eating window.” (lines 262-265)

Comment 4: With information about health status and health problems (e.g. CVD), have patients' rates improved? Has this been studied at all? E.g. measurement of blood pressure, blood lipid profile, glucose?

Response 4: We acknowledge that some clinical notes include subjective reports and health parameters such as lipid profiles and glucose levels. However, as this data was limited, we did not systematically analyse or include it. These limitations are acknowledged in the discussion: “Limited data on related parameters, such as lipid profiles and glucose levels, further restricted the scope of this study” (Lines 390–391). We also suggested this as an area for future research: “Future research should evaluate the long-term impacts of TRE, monitor adherence, address confounding variables alongside broader metabolic parameters, as well as validate RQ as a reliable tool for guiding dietary recommendations.” (Lines 395–397)

Comment 5: In Table 4, add the time of follow-up in the column (e.g., after 4, 6 months).

Response 5: We have included the median and interquartile range of the follow-up time for each group in Table 4 and clarified that the indicated numbers are in days (Table 4, Pages 8–9).

Comment 6: As economic status asked when collecting the interview? Because it may have deteriorated or improved, which affects the level of consumption of selected products (please add in text).

Response 6: Economic status data was not available in the clinic’s patient health records and was therefore not included in the analysis. However, we acknowledge that dietary intake changes could have influenced outcomes. This has been discussed: “Furthermore, the observed improvements may not be solely attributed to TRE, as factors such as dietary changes and increased physical activity which were not measured may have contributed.” (Lines 306-308).

Comment 7: The conclusions are very general, I would rather detail on a few points , especially since the authors mention improving metabolic health (that is who) and adding information of future interventions.

Response 7: We have added to the conclusions “While no formal guidelines currently incorporate TRE as a practice, practitioners might consider it as a complementary approach to enhance metabolic improvements, particularly when combined with currently recommended strategies such as healthier eating choices and exercise” (lines 408-412)

Comment 8: I have to admit that the work has potential, but I am very interested in this particular interval of ''fasting time'' because it raises a lot of controversy. I would like to ask you to clarify this, especially since from the title I expected just to be more detailed and better described.

Response 8: We have clarified the interval of fasting times in the results section, specifically detailing the fasting windows practiced (e.g., 16:8 and OMAD) and the rationale behind their recommendation in the clinical setting (lines 158-163 and lines 262 – 266). We acknowledge our limitation in describing this further as we are constrained by the available clinical records.

We sincerely appreciate your positive evaluation and your valuable input.

Best regards,

Authors

Reviewer 2 Report

Comments and Suggestions for Authors

here are some comments/questions/suggestions

paper is well written and interesting

would suggests to provide some description of the population of study, culture, general lifestyle? to better set the context of the study

statistical analyses seems adequate - however, could explain more the different variables/characteristics used in the univariate analyses - table 2

eg. does ethnicity matters in metabolism? lifestyle? provide some literature

although most variables are not significant - which in my opinion seems adequate - which means no matter of demographics effects of TRE seems to be the same

authors could also provide more practical implications or recommendations - so what now?

Author Response

Comment 1: Paper is well written and interesting. would suggests to provide some description of the population of study, culture, general lifestyle? to better set the context of the study

Response 1: Thank you for your suggestion. We have included a description of the population, culture, and general lifestyle in the methods section (Lines 72–78):

The data were sourced from two specialised metabolic health allied health clinics in Perth, Western Australia. These clinics utilised an ECAL® machine for indirect calorimetry testing to assess patient metabolism profiles. Clinicians were qualified Exercise Physiologists or Dietitians, who provided face-to-face services primarily focusing on weight loss and associated health conditions, with patients incurring out-of-pocket expenses for professional consultations and testing. Systematic data collection commenced in 2019 when the clinic transitioned to online data collection.”

Additionally, in the results section (Lines 214–224), we stated:

“Among patients in the Stage 1 Cohort, the mean age was 47.4 years (SD = 12.6), with the majority being female (n = 207, 76%) and of Caucasian ethnicity (90%)… A minority had specific dietary preferences prior to the clinic visit, such as following a ketogenic diet (11%) or being vegan/vegetarian (3%).

These points are further detailed in Table 1, which provides descriptions of ethnicity, employment status, dietary habits, and food preferences, reflecting general lifestyle characteristics.

Comment 2: Statistical analyses seems adequate - however, could explain more the different variables/characteristics used in the univariate analyses - table 2. eg. does ethnicity matters in metabolism? lifestyle? provide some literature

Response 2: Due to the homogeneity of our study population, with 90% of participants identifying as Caucasian, we did not perform analyses exploring the role of ethnicity in metabolism. This limited variation did not allow for meaningful statistical comparisons. We have acknowledged this limitation and suggested it for future research, as stated:

Further research with improved study designs is needed to confirm TRE’s acceptability and effectiveness across diverse patient populations” (Lines 400–402).

We did analyse some related lifestyle factors, such as skipping meals and breakfast habits, as these may influence whether patients adopt TRE. However, no significant associations were found. This point has been added to the discussion with additional references:

Further, patients are particularly drawn to TRE due to its flexibility, since it can be tailored to fit their personal lifestyle and preferences, making it a more sustainable option in the longer term for many individuals” (Lines 326–329).

Comment 3: Although most variables are not significant - which in my opinion seems adequate - which means no matter of demographics effects of TRE seems to be the same.

Response 3:  Thank you for this observation. We agree with your interpretation that the lack of significant differences among demographic variables suggests that TRE might be broadly suitable and generally acceptable across diverse patient groups. This has been included in the discussion lines 322-328.

Comment 4: Authors could also provide more practical implications or recommendations - so what now?

Response 4: We have added more specific practical implications and recommendations to the discussion section and conclusion:

Future research should evaluate the long-term impacts of TRE, monitor adherence, address confounding variables alongside broader metabolic parameters, as well as validate RQ as a reliable tool for guiding dietary recommendations. Exploring potential adverse effects, such as hunger or headaches and how to manage them could enhance TRE’s implementation and efficacy. While previous studies and a systematic review suggests generally high adherence to TRE, further research with improved study designs are needed to confirm TRE’s acceptability and effectiveness across diverse patient population.” (Lines 395–401).

While no formal guidelines currently incorporate TRE as a practice, practitioners might consider it as a complementary approach to enhance metabolic improvements, particularly when combined with currently recommended strategies such as healthier eating choices and exercise” (Lines 409–412).

We appreciate your constructive comments, which have significantly improved the clarity and quality of our manuscript.

Best regards,

Authors